# Green Biodegradable Polylactide-Based Polyurethane Triblock Copolymers Reinforced with Cellulose Nanowhiskers

**DOI:** 10.3390/jfb14030118

**Published:** 2023-02-21

**Authors:** Mohamed Khattab, Noha Abdel Hady, Yaser Dahman

**Affiliations:** Department of Chemical Engineering, Toronto Metropolitan University, Toronto, ON M5B 2K3, Canada

**Keywords:** biodegradable polyurethane, polylactide, drug delivery, tissue engineering scaffolds, polymer nanocomposites, triblock polyurethanes, bacterial cellulose nanowhiskers

## Abstract

A novel series of biodegradable polylactide-based triblock polyurethane (TBPU) copolymers covering a wide range of molecular weights and compositions were synthesized for potential use in biomedical applications. This new class of copolymers showed tailored mechanical properties, improved degradation rates, and enhanced cell attachment potential compared to polylactide homopolymer. Triblock copolymers, (TB) PL-PEG-PL, of different compositions were first synthesized from lactide and polyethylene glycol (PEG) via ring-opening polymerization in the presence of tin octoate as the catalyst. After which, polycaprolactone diol (PCL-diol) reacted with TB copolymers using 1,4-butane diisocyanate (BDI) as a nontoxic chain extender to form the final TBPUs. The final composition, molecular weight, thermal properties, hydrophilicity, and biodegradation rates of the obtained TB copolymers, and the corresponding TBPUs were characterized using ^1^H-NMR, GPC, FTIR, DSC, and SEM, and contact angle measurements. Results obtained from the lower molecular weight series of TBPUs demonstrated potential use in drug delivery and imaging contrast agents due to their high hydrophilicity and degradation rates. On the other hand, the higher molecular weight series of TBPUs exhibited improved hydrophilicity and degradation rates compared to PL-homopolymer. Moreover, they displayed improved tailored mechanical properties suitable for utilization as bone cement, or in regeneration medicinal applications of cartilage, trabecular, and cancellous bone implants. Furthermore, the polymer nanocomposites obtained by reinforcing the TBPU3 matrix with 7% (*w*/*w*) bacterial cellulose nanowhiskers (BCNW) displayed a ~16% increase in tensile strength, and 330% in % elongation compared with PL-homo polymer.

## 1. Introduction

Biodegradable polymers are increasingly used in different biomedical applications. This includes drug-releasing implants, bioresorbable surgical sutures, biodegradable vascular grafts, and in sustained drug delivery applications [1]. These polymers are chosen based on their native biocompatibility and biodegradation, but some often lack total biocompatibility, which has forced researchers to look for new polymers with improved properties. Polymers, such as poly(lactic acid) (PLA), poly(glycolic acid) (PGA), and poly(lactic-co-glycolide) (PLGA), are the most widely utilized since these are degradable by the ester bonds’ hydrolysis that leads to the formation of resorbable nontoxic degradation products [2]. However, their confronting drawbacks are some biocompatibility concerns due to the formation of acidic degradation by-products [3], small particles released during degradation that might trigger an inflammatory response [4], and the lack of cell attachment due to their high hydrophobicity [5]. With the current development in the biomedical field, there is a necessity to design and synthesize new polymers that are nontoxic and biodegradable with the ability to overcome the aforementioned drawbacks to expand their uses in regenerative medicine.

Segmented polyurethanes (SPU) are some of the best emerging solutions as they can easily be tailored to meet specific requirements, such as obtaining materials of specific physicochemical properties, improved mechanical strength, and controlled degradation rates [6]. However, these materials release toxic and carcinogenic byproducts as a consequence of harmful diisocyanates, which are often used as chain linkers. Accordingly, researchers replaced harmful chain linkers with safer products that do not produce toxic degradation byproducts [7]. Skarja and Woodhouse (2000 and 2001) have developed a class of novel biodegradable SPU that can be used in soft tissue applications. The hard segments of SPU are composed of L-lysine derived diisocyanate (LDI) and a phenylalanine diester as chain extender, whereas the soft segments are compiled poly(caprolactone) (PCL) or poly(ethylene oxide) (PEO) [8,9]. Hilborn et al. (2007) synthesized polyurethanes using LDI and biodegradable macrodiols (copolymers of trimethylene carbonate, 3-caprolactone, and D, _L_-lactic acid). These PUs are highly elastic with low degradation rates that makes them suitable for applications as long-term scaffolds [10]. Current studies on these biodegradable polyurethanes were not focusing solely on their utilization for long-term degradation applications, such as tissue engineering scaffolds, but also on their clinical significance for short-term application as rapid drug delivery system and magnetic resonance contrast agent in biomedical imaging (MRI) [11,12]. Amphiphilic polyurethanes block copolymer is a special class of biodegradable materials that can form core–shell micelles structure, which in turn could encapsulate a variety of drug molecules or magnetic nanoparticles inside their core cavities to enhance their biostability in vivo.

Polymer nanocomposites based on polyurethane and green nanofibers have attracted tremendous attention as a result of added nanofibers, and their substantial ability to enhance polymer properties, even at relatively low nanofiber loading [13]. Among natural nanofibers, Bacterial cellulose (BC) is considered one of the most abundant natural biocompatible nanofibers that is synthesized in the interior of the bacterial cells. BC nanofibers exhibited non-toxic effects on endothelial cells and minor effects on the blood profile. They have been approved by the FDA and extensively used as a starting material for many biomedical applications, such as wound dressings, biomimetic scaffolds, and drug delivery devices [14]. In the current study, bacterial cellulose nanowhiskers (BCNW), obtained by controlled acidic hydrolysis of BC nanofibers, are used as green and natural reinforcing nanofibers. Due to its high aspect ratio, (length/diameter) of 30 to 150, hydrophilicity, high crystallinity, and excellent mechanical properties, BCNW are considered an excellent candidate for reinforcing polyurethane matrix, even at low loading [15].

This project is an attempt to overcome the prevailing limitations confronting the use of biodegradable homopolymers by focusing on the synthesis of nontoxic biodegradable triblock polyurethanes (TBPUs) of different segments and molecular weights. The structure of these designated polyurethanes is based on triblock copolymers (TB) that are composed of polylactide and polyethylene glycol (PL-PEG-PL) as hard segments and polycaprolactone (PCL) as soft segments. The newly synthesized TBPUs are expected to have improved hydrophilicity, biodegradation rate, and cell attachment abilities over their corresponding pure homopolymers, i.e., PLA and PCL. During the synthesis of the triblock copolymer, PEG was introduced in different ratios during polymerization reaction to enhance the hydrophilicity and degradability of PL segments [16], whereas PCL segments are used to improve flexibility and elongation. In addition, the nontoxic 1,4-butane diisocyanate (BDI) was used as a chain linker to avoid the release of toxic and carcinogenic byproduct after degradation. The use of BDI was of special interest since, upon degradation, it yields 1,4-butane diamine “known as putrescine”, which is already present in mammalian cells [17]. This is highly beneficial when considering the biocompatibility of the prepared poly(ester-urethanes) for biomedical applications. Moreover, the envisioned BCNW/TBPUs nanocomposites scaffolds are expected from one side to have an improved interfacial adhesion with the newly generated cells due to the presence of either hydrophilic BCNW or PEG segments in the hydrophobic polymer matrix. On the other side, they will improve the mechanical strength and biodegradation rate of polymer nanocomposites. This project is a multifaceted challenge, since obtaining a new group of materials with a given set of mechanical and physical properties to be utilized in biomedical applications is conditional upon being biocompatible and biodegradable.

## 2. Experimental

### 2.1. Materials

All chemicals were purchased from Sigma-Aldrich and were used as received unless stated otherwise. Lactide [M_n_ = 144.13 g/mol], polyethylene glycol as a linker [PEG; M_n_ = 4000 g/mol], and Sn(Oct)_2_ as a catalyst were used to synthesize triblock copolymer. A total 1,4 Butane diisocyanate [BDI; M_n_ = 140.14 g/mol] and polycaprolactone diol [PCL-diol; M_n_ = 2000 g/mol] soft segments were used in chain extending reactions, and the formation of final polyurethane matrix. Diethyl ether, dimethylformamide (DMF), dichloromethane (DCM, 99.5%), N,N-dimethylacetamide (DMAc), acetone, and methanol of ACS grade were used in the present work as common solvents during synthesis, purification, and film casting of polyurethane. Deuterated chloroform (CDCl_3_, 99.8%) was used for NMR analysis. Phosphate buffered saline (PBS, pH 7.4), sodium azide NaN_3_, and porcine pancreas lipase type II were obtained from Sigma-Aldrich Canada, (lipase activity: 100–500 units/mg protein (using olive oil (30 min incubation)), 30–90 units/mg protein (using triacetin)). Sulfuric acid (98%) was used for the hydrolysis of bacterial cellulose nanofibers to obtain the nanowhiskers.

### 2.2. Experimental Procedure

#### 2.2.1. Production of BC Nanofibers

The production of bacterial cellulose nanofibers was carried out under shaking conditions in 500 mL flasks containing 200 mL of the fermentation medium [18,19]. The media composition was as follows: 40 g/L Fructose (carbon source), 5 mL of corn steep liquor (CSL; nitrogen source), 0.25 g/L of MgSO_4_.7H_2_O, 1 g/L of KH_2_PO_4_, 2.42 mg/L of Sodium molybdate (NaMoO_4_.2H_2_O), 3.6 mg/L of FeSO_4_.7H_2_O, 3.3 g/L of (NH_4_)_2_SO_4_, 14.7 mg/L of CaCl_2_.2H_2_O, 1.73 mg/L of ZnSO_4_.7H_2_O, 0.05 mg/L of CuSO_4_.5H_2_O, 1.39 mg/L of MnSO_4_.5H_2_O, 2 mg/L of Inositol, 0.4 mg/L of Nicotinic Acid, 0.2 mg/L of D-Pantothenic Acid 0.2 mg/L of Riboflavin, 0.2 μg/L of D-Biotin, 0.4 g/L of Thiamine Hydrochloride, 0.4 mg/L of Pyridoxine Hydrochloride, and 0.2 g/L of Folic Acid. All glassware was sterilized in an autoclave (Sanyo MLS 3780; Thermo Scientific, Toronto, ON, Canada) at 121 °C for 20 min prior to use. Carbohydrate solution and its additives were adjusted at initial pH 5.0. Then, the mixture was sterilized separately from CSL at 121 °C for 20 min to avoid Maillard reaction. CSL was added to the growth medium under aseptic conditions and, if necessary, sterilized distilled water was added to compensate for evaporated water during autoclaving. Each flask was aseptically inoculated using 2 mL of the inoculums after being allowed to cool down to room temperature. Then, the flask was incubated at 29 °C for 7 days with shaking speed of 250 rpm (MaxQ 2000; Thermo Scientific, Toronto, Canada). The pH of each flask was checked after 7 days, and cell lyses of solutions was performed with excess 2 M NaOH at 100 °C for 15 min in the autoclave. BC nanofibers were extracted and repeatedly washed with distilled water. Production of BC was quantified gravimetrically based on the dry weight of the BC obtained. A production of 46 g/L on a wet basis was achieved from the stirred culture.

#### 2.2.2. Preparation of BCNW

BC pellicles were ground in a blender and the gel-like material was then squeezed in order to remove most of the absorbed water. The preparation procedure of BCNW was similar to that reported earlier [20]. Accordingly, hydrolysis of the dried BC was performed with 60 wt% sulfuric acid under stirring for 2–3 h at 50 °C until a homogeneous solution was obtained. The acid/BC ratio was kept constant at approximately 70 mL/g. After that, the cellulose nanowhiskers were obtained by centrifugation as a white precipitate, then neutralized with sodium hydroxide until a neutral pH was obtained. Subsequently, BCNW was re-suspended and washed by deionized water using several centrifugation cycles, and finally obtained as a partially hydrated precipitate. BCNW was solvent exchanged into N,N-dimethylacetamide (DMAc), where water in the partially hydrated precipitate was replaced by DMAc by applying several centrifugation cycles in which the supernatant was removed and replaced with fresh DMAc several times. After that, dichloromethane (DCM) was added to DMAc–whiskers solution and refluxed for 2 h at 80 °C with contentious stirring until BCNW became well dispersed in DMAc/DCM mixture. A 2.2 wt% of BCNW was obtained after partially evaporating the solvent.

#### 2.2.3. Synthesis of PL-PEG-PL Triblock Copolymer (TB)

The bulk polymerization of lactide was initiated by the hydroxyl moiety of PEG according to Leenslag and Pennings’ method with minor modifications [21]. Prescribed amounts of an initiator PEG and lactide were uniformly mixed as the preplanned feed weight ratio shown in Table 1. The reaction mixture was placed in a three-necked reactor flask equipped with an overhead mechanical stirring shaft, a reflux condenser, and a nitrogen gas inlet. The mixture was directly evacuated, then dehydrated for 1 h under reduced pressure at 120 °C. After LA and PEG have been melted, the system was purged with N_2_ gas and the catalyst in chloroform solution was added according to the prescribed weight percent ratio of dehydrated reactants. The polymerization reaction was continued for 24 h, where the temperature of the oil bath in which the reactor was immersed was kept at 140 °C. As the polymerization reaction precedes, the reactants’ mixture becomes less transparent and more viscous. At the end of the reaction the product was annealed at 135 °C for about 180 min to make sure no residual monomer was left over. The reaction vessel was partially cooled down, and a small amount of chloroform solution was added to extract the product from the reaction vessel before solidification. Six different TB copolymers were synthesized by varying the initial feed ratios of LA to PEG. As a comparison reference, homo-PLs were also synthesized under similar conditions as previously mentioned without adding PEG.

#### 2.2.4. Purification and Recovery of TB Copolymers

The higher molecular weight members of TB copolymer samples were purified. Each sample was initially dissolved in chloroform, then methanol is slowly added to the solution with continuous stirring at 30 °C until the solution became turbid. The solution was aged for about 2 h at this temperature for complete precipitation where the viscous polymer was separated by decantation. Whereas the lower molecular weight members of TB were precipitated as white powder from chloroform solution into ether, then separated by vacuum filtration. The products were dried in desiccators for at least 24 h at room temperature prior to further use.

#### 2.2.5. Synthesis of TB-BDI Pre-Polymer

The TB-BDI pre-polymer was synthesized by mixing the prescribed amount of TB copolymer with a stannous octoate catalyst in a three-necked reaction flask equipped with overhead mechanical stirring under nitrogen atmosphere. The BDI linker dissolved in the least amount of chloroform, which was added to TB in a 2:1 molar ratio. The reaction mixture was heated in an oil bath for 2 h at 70 °C. After the reaction, the product was precipitated in excess methanol, decanted, and dried in a desiccator for 24 h at room temperature.

#### 2.2.6. Synthesis of Triblock Polyurethane Polymers (TBPUs)

Six different TBPU polymers were obtained with varying compositions. A chain-extension was carried out to produce high molecular weight TBPU polymers as follows: PCL diol was used as a soft segment. The PCL content is based on balancing the isocyanate and PCL diol concentration to a 1:1 mole ratio. The reaction vessel was kept under nitrogen blanket at 70 °C for 4 h. Higher molecular weight members were precipitated in methanol, whereas the members of lower M_n_ were precipitated in diethyl ether. The precipitated product was filtered, rinsed with methanol, then dried overnight in a vacuum oven at 40 °C.

#### 2.2.7. Preparation of Polymer Films

Polymers solutions, 15 wt% in DMAc, were prepared by stirring the polymer/solvent mixture in an oil bath at 80 °C. Polymer films of dimensions (1 mm thickness × 1 cm width × 10 cm length) were fabricated by casting into handmade stainless-steel molds. Initially, solvent evaporation was performed at room temperature. Films were removed from the mold and dried under vacuum condition at 40 °C for 24 h to ensure complete solvent evaporation.

#### 2.2.8. Preparation of TBPU/BCNW Nanocomposites

The TBPU/BCNW polymer nanocomposites were prepared by the solvent casting method as reported earlier in the literature, but with different solvents [22]. First, TBPU polymer solution was prepared by dissolving the polymer in DMAc/DCM mixture at 70°C. Polymer solution was gradually added with continuous stirring into the suspension of cellulose nanowhiskers dispersed in DMAc/DCM with amounts ratio of (1, 3, 5, 7, and 8 w% based on polymer content, respectively). After which, the polymer nanocomposites mixture was poured into a mold, then flashed frozen in liquid nitrogen. The mold was then transferred to a freezer set at −50 °C and kept for 24 h. After that, final nanocomposites were kept at 40 °C, and under vacuum condition for 48 h to ensure complete solvent evaporation.

### 2.3. Characterization Techniques

#### 2.3.1. Fourier Transform Infra-Red Spectroscopy (FTIR/ATR)

Structural changes during the stepwise formation of TBPU polymers were investigated by FTIR spectroscopy using Perkin Elmer Spectrum-1 instrument (Waltham, MA, USA) in attenuated total reflectance mode (ATR). The ATR spectra of all samples were recorded as transmittance in the range of 4000–500 cm^−1^. The ATR-crystal used was ZnSe, and each spectrum was recorded with resolution of 4 cm^−1^ and consisted of 20 scans.

#### 2.3.2. H-NMR Spectra

The chemical compositions of polyurethanes and their TB copolymers precursors were characterized by recording ^1^H-NMR spectra using Bruker 400 MHz Spectrometer Biospin (Rheinstetten, Germany) located at Ryerson University Analytical Centre (RUAC). The polymers were dissolved in CDCl_3_-d^1^ at a concentration level of 10 mg/mL, and tetramethylsilane was used as an internal reference at 25 °C. The degree of polymerization (DP) of PL in all copolymer samples was calculated from ^1^H-NMR spectra [23,24].

#### 2.3.3. Gel Permeation Chromatography (GPC)

The number-average molecular weights (Mn) of PL-PEG-PL TB copolymer were determined using a Viscotek GPC/SEC system (Westborough, MA, USA). The system was equipped with a Triple detector array (TDA 302) that can give very accurate dn/dc determinations and, subsequently, molecular weight. This consists of Right-Angle Light Scattering (RALS), a high sensitivity Viscometer (for DP and IP), and a Refractive index detector (RI). Spectra analysis and data collection was conducted using the OmniSec 5.1 software. Tetrahydrofuran solvent (HPLC grade) was used as Eluent with a flow rate of 1.0 mL min^−l^ at 32 °C through the Shodex GPC KF-802 series column (Tokyo, Japan). Polymer samples were dissolved in THF at a concentration level of 10 mg/mL, then filtered through a 0.45μm filter. The GPC/SEC was calibrated using polystyrene reference samples having narrow molecular weight distributions ranging from 1260 g/mol to 184,900 g/mol.

#### 2.3.4. Differential Scanning Calorimetry (DSC)

A differential scanning calorimeter (DSC) was used to measure the thermal properties of the polymers. The Perkin Elmer Diamond Differential Scanning Calorimeter controlled with PYRIS 7 software was used. The thermograms covered a temperature range −20–200 °C under the nitrogen atmosphere at a flow rate of 20 mL/min and heating rate of 10 °C min^−1^. Approximately 5 mg polymer samples were placed and sealed in aluminum pan (20 μL). The first scan measured the melting endotherm, and the second measured T_g_ values.

#### 2.3.5. Water Content Measurements

Water contents of the polymer samples were determined by soaking the samples in deionized water at 25 °C. Samples were collected for weighing after being gently blotted until equilibrium was achieved. Water content is expressed as a percentage of dry polymer samples and calculated using Equation (1). The recorded water contents were taken as an average of at least two determinations.
% Water content = (W_swollen_ − W_dried_)/W_dried_ × 100% (1)
where W_swollen_ and W_dried_ are the weights of the swollen and dried polymer, respectively.

#### 2.3.6. Contact Angle Measurements

Water in air contact angle measurements were performed on different TBPUs films on an Optical-Bench Contact Angle Goniometer (Hamburg, Germany). Each reported value was taken as an average of at least three measurements. A droplet of distilled water was deposited on the samples and the contact angle was measured at different times.

#### 2.3.7. Biodegradation

In this study, hydrolytic and enzymatic degradation were employed for all the synthesized polyurethane samples. Hydrolytic degradation was carried out in PBS solution. This solution was composed of 0.1 M PBS with 0.9% NaCl, 0.02% NaN_3_, and pH 7.4. Enzymatic degradation was conducted in the same PBS solution of the same pH but containing 0.1 mg mL^−1^ Lipase from porcine pancreas. Each sample was placed into an individual vial containing 10 mL PBS/enzyme mixture, then incubated with shaking at 37 °C to simulate in vivo dynamic tissue environment. The samples were taken out after 5 h, 15 h, 30 h, 60 h, and 120 h, rinsed by deionized water, vacuum dried at 60 °C for 24 h, and reweighed for weight loss determination. The reported weight loss was calculated using Equation (2) and taken as an average from three samples.
Weight loss (%) = (W_0_ − W_t_)/W_0_ × 100%(2)
where W_0_ and W_t_ are the dry weight of the sample before and after degradation, respectively.

The pH measurements were collected for the degradation solutions. Scanning electron microscope (SEM) model JSM-6380 LV (Oxford, UK) with a monochromator (Al X-ray source) operated between 5–20 kV was then used to examine the surface morphology of polymer samples after were gold coated.

#### 2.3.8. Field Emission Scanning Electron Microscopy (FE-SEM)

The surface morphology of PUs samples before and after degradation was examined using FE-SEM, FEI Quanta 200 F microscope (Hillsboro, OR, USA) with an accelerating voltage of 15–20 kV. Gold surface layer was sprayed on the polymer samples by an ion sputter coater with a low deposition rate prior to being examined.

#### 2.3.9. Mechanical Testing

The mechanical properties of the TB, TBPU, and TBPU/BCNW nanocomposites were evaluated by measuring tensile strength, tensile modulus, and elongation at break. Labthink’s Param XLW (PC) Auto Tensile Tester (Jinan, China) was used for mechanical testing. It was equipped with a 500 N load cell, and operating at room temperature and a crosshead speed of 100 mm/min. Most of the measurements were conducted in triplicates within an average total error not exceeding 5%.

## 3. Results and Discussion

### 3.1. Synthesis of Triblock (PL-PEG-PL) and PUs

A series of triblock polyurethane consisting of PL, PEG, and PCL were prepared via three-step polymerization reactions (Figure 1). In the first reaction, the triblock copolymers PL-PEG-PL were synthesized through the ring opening polymerization of lactide (LA) in the presence of bifunctional macro-monomer dihydroxy PEG as initiator and stannous octoate as the catalyst. The molecular weights of the triblock PL-PEG-PL were controlled by changing the feed ratio of lactide and PEG. The %*w*/*w* feed ratios of LA/PEG, reaction yield, and reaction conditions along with observations are summarized in Table 1. For convenience, six triblock polymer samples were prepared and named from TB1 to TB6. The TB copolymers of lower molecular weight and with higher PEG content showed noticeable solubility in water.

In the second step, TB-BDI pre-polymer was synthesized through the condensation reaction between the previously synthesized TB and BDI in 1:2 molar ratio in the presence of Sn(Oct)_2_. Following the second step, the chain-extension reaction was carried out by reacting PCL-diol (the flexible segment) with TB-BDI in a 2:1 molar ratio to form the final triblock polyurethane polymers TBPUs. To control the final molecular weight, and to examine the effect of PL/PEG segment ratio on the final physical properties of the TBPUs polymers, the initial molar ratios of TB: PCL diol: BDI were kept consistent at 1:2:2. For convenience, the obtained polyurethane samples are labeled from TBPU-1 to TBPU-6.

### 3.2. Characterization of PL-PEG-PL and PUs

Figure 1 shows the stepwise formation of the triblock copolymers (PL-PEG-PL) from LA and PEG precursors as confirmed by FTIR in ATR mode. As shown in the figure, the lactide ester carbonyl band appeared at around 1750 cm^−1^, and the C–H stretching band of CH_2_ group in PEG appeared at 2880 cm^−1^ [25]. In comparison with LA and PEG homopolymers absorption bands, the following band assignments arose in the triblock copolymer spectrum: the band at 2995 cm^−1^ belonged to C-H stretching of -CH_3_ of LA units; the bands at 2865 cm^−1^ are a result of C-H stretching of -CH_2_ groups of PEG; the band at 1750 cm^−1^ is a result of C=O stretching of the LA units and the bands at 1095 cm^−1^ are a result of C-O stretching of LA and the ether bond of PEG. Hence, the appearance of all the characteristic absorption bands that belong to LA and PEG in the spectrum of the triblock copolymer confirms the formation of the triblock copolymer PL-PEG-PL.

The FTIR spectra of TBPU-1 in Figure 2 displays the formation of polyurethanes from TB-BDI and PCL-diol. In addition to the characteristic peaks that are present in TB copolymer, FTIR spectra of TBPU-1 revealed new characteristic absorption bands at 2945, 1182, and 1485 cm^−1^, which ascribed to C-H, C-O, and C–N stretching [26], respectively. In addition, the characteristic broad peak due to hydroxyl group stretching in the TB copolymer disappeared due to urethane condensation reaction of TB copolymer with BDI, while new peaks arose at 3473 cm^−1^ due to –C=O–NH– stretching in secondary amide, and –N=C=O stretching at 2287 cm^−1^ due to isocyanate [27]. After the addition of the chain extender PCL-diol, the isocyanate peaks disappeared, while the peak due to –C=O–NH– stretching at 3473 cm^−1^ was retained, which implies the successful preparation of TBPU polymers.

The chemical shifts of the various hydrogen atoms in the copolymer were determined by ^1^H-NMR spectra as shown in Figure 3. As a common fixture in all ^1^H-NMR of the prepared TB copolymers (i.e., from TB1 to TB6), the signals at 5.1 ppm and 1.55 belong to the protons of (-CH-O-) and (CH_3_) groups of PL blocks. Whereas the signal at 3.64 ppm is characteristic for the main chain methylene units -(CH_2_)- in the PEG blocks. Additionally, a small but highly significant signal of the methylene protons of PL-connecting ethylene glycol units (-CH-COO-CH_2_-) also appeared at 4.25 ppm. The obtained ^1^H-NMR signals were a good match with the reported literature values for PEG/PL copolymers of different blocks [28,29]. In all spectra, the “extra peaks” determined at the chemical shifts (δ = 1.85 ppm,m) and (δ = 2.175 ppm,s) were solvent residual peaks due to the interaction of CH_2_ of tetrahydrofuran and CH_3_ of acetone with CDCl_3_ [30]. Considering all the previous findings, it was concluded that covalently bonded block copolymers that are comprised of PEG and PL sequences were successfully synthesized.

In addition, the molecular weights, average segmental length of both LA and PEG components, and the composition of the copolymers were also estimated from ^1^H-NMR spectra. This was accomplished by comparing the protons peaks in the methylene groups of PEG segments (b) with the lactoyl methane protons peaks of the PL segments (d), which are centered at 3.64 ppm and 5.2 ppm, respectively; see Figure 3. As evidenced from the obtained ^1^H-NMR spectra of the prepared polymers, the relative area of proton peaks at (b) and (d) has varied according to the initial amount of feed ratio of LA/PEG monomers. The peak area ratio that was obtained from integration values of CH_2_ peaks for EG and CH for PL was used to determine % content and block ratio of EG in the copolymers according to Equations (3) and (4).
% content of EG in triblock copolymer = {(Y/4)/[(Y/4) + (X/2)] × 100}(3)
Block ratio = the number of EG blocks/the number of LA blocks = [(Y/(2 × DP_PL_)/X/(4 × DP_PEG_)] (4)
where Y is the integration value for the EG ethylene units’ peak, and X is that of LA’s CH peak. Moreover, the degree of polymerization of PL and the segment length of the PL in the block copolymer were also estimated from the ^1^H-NMR spectra based on the peak intensity ratio of the methylene protons of EG (i.e., OCH_2_CH_2_: δ = 3.64 ppm) and the methine proton of the LA unit (i.e., COCH(CH_3_): δ = 5.2 ppm). For instance, considering the M_n_ of PEG 4000 g/mole, the degree of polymerization of PL can be calculated from the relation; DP_PL_ = DP_PEG_ × (2X_LA_/Y_EG_). Ultimately, the M_n_ of the PL segments determined was found in the range of 6256−66,682 g/mole. Hence, the molecular weight of the triblock copolymer can then be estimated from the relation M_n(TB)_ = 144[DP_PL_] + [M_n(PEG)_ − 18]. The obtained M_n_ values of the triblock PL-PEG-PL were found in the range of (16,494–137,343 g/mole). These values were a good match with the calculated value based on GPC measurements; see Table 2.

As presented in Table 2, within the given range of the molar ratio of PEG added, the molecular weights of the triblock copolymers were found to be inversely proportional to the amount of added PEG in the feed. This may be a result of the decrease in DP of PL as PEG content increased. Therefore, it is apparent that adjusting the block length of the PL in the triblock PL-PEG-PL by changing the amount of PEG macro-monomer in the feed is possible, and may lead to further control over the final physicochemical properties of the TBPUs.

The structures of the final TBPUs polymer that were produced as a result of the interaction of PCL with triblock PL-PEG-PL were further confirmed using ^1^H-NMR spectra. In this regard, besides the predetermined characteristic signals of the triblock PL-PEG-PL, new chemical shift signals emerged due to the presence of PCL segments; see Figure 4. As shown in the figure, the urethane bond signals are not seen as well as they partly overlapped with the solvent peak at 7.23 ppm. However, the formation of the urethane bond peak in this region was further confirmed with the use of deuterated acetone as a solvent, and was previously confirmed by FTIR.

After the coupling reaction between TB copolymer and BDI molecules, ^1^H-NMR spectra for all samples revealed neither remarkable change happened in the PL/PEG segment ratio or an increase in the molecular weight of the polymer. This result evidenced that BDI molecules interact only with the terminal OH groups without causing any crosslinking reactions between TB copolymer chains. Therefore, based on the controlled molar ratio of the terminal OH groups of PCL-diol, and the molar ratio of added BDI, (i.e., 1:2:2; TB/PCL-diol/BDI), the number average molecular weight of TBPUs can be estimated using the equation:M_n (TBPU)_ = M_n (TB)_ + [2 × MW_BDI]_ + [(2 × M_n PCL-diol)_] (5)

The M_n_ values of TBPUs were found in the range of (18,779–139,769 g/mol), see Table 3.

### 3.3. Water Absorption and Contact Angle Testing of TBPUs

The surface and bulk hydrophilicity of the various TBPU copolymers was determined by water uptake and contact angle measurements. The preliminary lab observations revealed that when the specimens made of homo-PL and TBPU copolymers were immersed in water, the homo-PL specimen floated on the water due to its hydrophobic nature, whereas the TBPU specimens absorbed water rapidly and sank. These observations elucidated that synthesized TBPU copolymers hold more hydrophobic characters than the PL homo polymer. As can be seen in Figure 5, the water uptake of homo-PL did not exceed 3%, while those of the TBPU copolymers were above 16%. As shown in the figure, increasing the PEG content of the TBPU copolymers would enhance the water uptake and consequently lead to an improvement of their biodegradation. This observation explains why the degradation of hydrophobic PL is extremely slow with more than 80% of the original mass of implant remaining at the implantation site even after 6 months [31].

Water contact angle testing reports that better hydrophilicity results in a smaller contact angle [32]. The variations in hydrophilicity as a function of composition were assessed by observing the water contact angle, see Figure 6. The figure reveals that the observed contact angle decreases with increasing PEG molar ratio of TBPUs. Compared with homo polylactide, the trend of hydrophilicity improvement increases from TBPU-1 to TBPU-3 due to the increased content of the hydrophilic PEG segments. The contact angle measurements reveal that TBPUs copolymer exhibited markedly enhanced hydrophilicity when compared to polylactide homopolymer. For instance, TBPU-3 attained substantially larger equilibrium water contents around 31% compared to 3% for homo-PL. The increasing of PEG content from TBPU-1 to TBPU-3 readily explains the behavior encountered in Figure 6. Therefore, as it was the aim of this study, the hydrophilic modification of PL by introducing PEG during ring opening polymerization of LA showed success in providing control over hydrophobicity of homo polylactide. This improvement in the hydrophilicity of the high molecular weight TBPUs might find a good application for improving the cell attachment abilities of TBPU scaffolds. Moreover, it also enhances the degradation behavior of the low molecular weight TBPUs when used for drug delivery applications. Therefore, the synthesized TBPUs can be considered as promising biodegradable future candidate for utilization in a biological environment.

It is worth mentioning that the contact angle measurements of TBPU samples with higher PEG contents (i.e., TBPU-4 to TBPU-6) could not be tested. Their higher hydrophilicity and partial solubility in water caused instant spreading of the water drop and sometimes local distortion on the polymer surface.

The variations in hydrophilicity as a function of composition and time were further assessed by observing the variation of the water contact angle with time as presented in Figure 6. The variation of the contact angles with time was observed for each polymer until [θ] = 30° was attained, which implies good surface wetting. Results summarized in Table 3 show that both initial contact angle and t_30_ increase with increased hydrophobicity, i.e., for the lower PEG content or the longer PL segments in TBPU copolymers. Figure 6 also demonstrates that, not only is the initial contact angle is much higher for homo-PL than the TBPU copolymers, but also the rate of change is higher, which is indicated from the significant decrease in the slope of the curves.

### 3.4. Degradation and Associated Morphological Changes

Many contributing factors are affecting the biodegradation of polyurethane polymers, such as hydrophilicity of polymer segments, the molecular weight of polymers, degree of microphase separation, and enzyme function. In this study, Lipase from the porcine pancreas was employed in the enzymatic degradation testing due to its proven ability to degrade PLA [33]. Enzymatic degradation data summarized in Table 3 showed that weight loss of 75%, 82%, and 91% were observed for TBPU-4, TBPU-5, and TBPU-6, respectively, which are rich in PEG segments. The degradation profiles displayed rapid enzymatic degradations for TBPU samples within 120 h. The initial weight loss increased rapidly in the first 15 h, then slowed down when degradation time of 60 h was reached. This high rate of degradation is probably attributed to the hydrolysis of the low molecular weight PL segments of polyurethane polymeric chains, which simultaneously is supported by the high content of the hydrophilic PEG segments (~47–52%). These together resulted in a low polymeric chain-chain interaction and facilitated water and enzyme attack to the polyurethane chains causing them to swell and dissolve easily in aqueous solution. The dissolution of water-soluble PEG segments is also a contributing factor in the weight loss of TBPUs along with enzymatic degradation. Moreover, the higher solubility of the leftover fragments that formed after polymer hydrolysis can also be considered as a good indication for reduced inflammatory response in the vicinity of the implanted tissues. This will prioritize the utilization of these materials in drug delivery systems.

In contrast, weight loss values of 21%, 23%, and 27%, respectively, were achieved after 5 days for the high molecular weight polyurethanes (i.e., TBPU-1, TBPU-2, and, TBPU-3); see Table 3. Both high phase separation and higher M_n_ lead to low degradation rate for polyurethane due to enhanced interaction force among the polymer chains that in turn resists water and enzyme attack [34]. This explains the slower enzymatic degradation rate of the synthesized high molecular weight polyurethanes than the other samples in the series. The high content and large molecular weight of hydrophobic PL segments that are present in those polymer chains also contribute to a slower enzymatic degradation rate.

Comparing the enzymatic degradation rate of TBPU-3 to and the homo polylactide polymer of comparable molecular weight (M_n_ = 67,500 g/mol; enzymatic degradation rate ~10%), one can conclude that the enzymatic degradation rate of TBPUs was approximately two times faster than homo-PL. This implies that the complete degradation of TBPU sample needs half the degradation time required for a homo-PL sample of comparable molecular weight.

To further investigate the effect of enzymatic action on degradation rate, a comparative hydrolytic degradation test was carried out for TBPU samples in PBS at pH 7.4 and 37 °C. Degradation data summarized in Table 3 shows that the hydrolytic degradation rates for all TBPU samples are much less than the corresponding values obtained from enzymatic degradation. This in turn verifies that the major cause for accelerated hydrolysis for polyurethanes is the attack on ester linkages of PL hard segments by the enzyme rather than the dissolution of the soft PEG segments.

Among the promising TBPU samples, TBPU-3, which had a relatively high M_n_ and good degradation rate, was selected as a scaffold representative and tested for full degradation. The lab observation showed that the full degradation of TBPU-3 in PBS solution (pH 7.4) needs approximately 1.5 months compared to 6.5 months required for the homo-PL sample. Nevertheless, a slight decrease of 0.5 pH units was observed at the end of depredation due to the release of some acidic byproducts from PL segments upon degradation. The non-significant change in pH associated with the weight loss of TBPUs suggests a reduced inflammatory response in vivo and supports the suitability of these materials for utilization as internal implants [35].

Moreover, the effect of changing pH value on hydrolytic degradation of TBPU-3 was also examined by studying the degradation profiles of TBPU-3 in PBS at different pH values; 7.4, 6.0, and 5.0. The degradation rate was found to be faster in PBS solution of pH 7.4 than the other acidic pHs. One probable reason is that pH 7.4 is a mildly basic medium, which promotes the hydrolysis of ester linkage by providing OH^−^ anions that act as strong nucleophiles [36]. The other reason is that the degradation of TBPUs produces acidic byproducts containing COOH groups, which are more soluble in basic medium and enhance the hydrolysis by shifting the equilibrium reaction to the forward direction.

The surface morphology changes associated with the degradation process of TBPU-3 films were monitored using SEM; see Figure 7. The surface topography of the polymer film before enzymatic or hydrolytic degradation showed smooth morphology (Figure 7A). After soaking the polymer film for 120 h in the Lipase enzyme, the surface becomes markedly eroded and holes and cracks appeared (Figure 7B). However, for non-enzymatic degradation in PBS at pH7.4, a similar pattern is observed, but is less harsh and slower than the enzymatic degradation (Figure 7C). The morphological changes observed by SEM suggest that TBPU can be slowly degraded without enzymatic action. This observation is in good agreement with the results obtained from weight loss experiments.

### 3.5. Thermal Properties of TBPUs Copolymers

The DSC thermograms of TBPU copolymers, neat PEG, and PL homopolymers are shown in Figure 8. As displayed in the figure, neat PEG and PL homo polymer showed T_m_ peaks at 64 °C and 186 °C, respectively. The thermograms of TBPUs copolymers of lower PEG content 10–18% (i.e., TBPU1–TBPU3) showed T_m_ peaks for PL segments only at slightly lower values than the homo-PL. More specifically, T_m_ values due to PL segments decreased further as PEG content increased. The absence of melting endotherm peaks of the PEG segment in those three TBPUs might be related to the lake of highly ordered and long crystalline domains for PEG soft segments compared to the long enough PL hard segments. However, the thermograms of TBPUs that have higher PEG contents 48–53% (i.e., TBPU4–TBPU6) showed broad T_m_ peaks at temperatures slightly higher than neat PEG, see Figure 8. The increase in the T_m_ values of PEG might be related to the partial obstruction of polymer chain mobility in the crystallization domain of PEG by the long PL segments. Since the T_m_ of both neat PEG and PL segments had altered one another, it can be used as an indication for a successful copolymerization. Nevertheless, no T_g_ peaks were observed in DSC curves of TBPUs related to hard or soft segments. A probable explanation is that copolymerization reaction may have caused a decrease in the structure symmetry, which significantly inhibited the chain packing [37]. Furthermore, in the second cooling run of TBPU-5, the T_m_ peaks corresponding to PEG segments disappeared whereas the T_m_ peaks for PL segments remained, see Figure 8. These observations showed that PL blocks readily crystallize with greater tendencies than PEG blocks. In turn, this limits the mobility of the PEG segments, and severely hinders the crystallization of the PEG by the already solidified PL segments.

Moreover, T_m_ of the soft PCL segments was not clearly observed in TBPUs isotherms. This might be related to the low M_n_ PCL used, which did not allow the formation of enough crystalline domains or because of the overlap of its endotherm in the melting range of PEG segments.

### 3.6. Mechanical Properties of TB and TBPUs copolymers

The mechanical properties of TB copolymers and their corresponding TBPUs were tested in this study by evaluating elongation at break (ε_b_), tensile strength at break (σ_b_), and tensile modulus (E), see Table 4. For the intended use of biodegradable PL in tissue engineering applications, a typical goal is to increase elongation at break and tensile toughness without adversely affecting the tensile strength and tensile modulus. Although it was possible to cast the TB copolymers and their corresponding TBPUs that have high PEG content and shorter PL segments into films, their mechanical strength was not high enough to be tested. Therefore, in this study, it was very challenging to obtain TBPU copolymer of both high mechanical strength and hydrophilicity.

Table 4 reveals that for the three highest molecular weight TB copolymers and their corresponding TBPUs, all mechanical properties (i.e., ε_b_, σ_b_, and E) were increased as the chain length of PL increased, and PEG content decreased. Due to the impressive mechanical properties, hydrophilicity, and good degradation behavior, TBPU-3 was chosen among the three highest molecular weight TBPUs as a pilot for comparison and fabrication of nanocomposite.

The tensile strength and elongation of PL, TB3, and TBPU-3 were examined at room temperature, and depicted in Figure 9. As shown in the figure, the stress–strain curve of the polylactide homo polymer demonstrates a typical rigidity and brittleness because of its high tensile modulus and tensile strength, but it’s very limited elongation at break. Compared with homo-PL, the presence of PEG segments in TB3 copolymer chains made it less brittle because it caused a decrease in tensile strength; however, both tensile modulus and elongation at break were interestingly increased. TBPU-3 showed similar behavior to TB3 except for a significant increase in elongation at the break that was observed and might be related to the presence of soft PCL segments that were added after the urethane reaction. As presented in Table 4, TBPU-3 exhibited a remarkable tensile strength of about 27.0 MPa and with a considerable elongation of 273% over the homo-PL. These mechanical properties are comparable to those of cartilage, trabecular, and cancellous bones and made TBPU-3 a promising candidate for utilization in tissue engineering scaffolds [38].

Figure 10 examines the effect of changing BCNW loading on both tensile strength and elongation at break for TBPU-3 nanocomposite. As displayed in the figure, an improvement in mechanical properties was obtained upon increasing BCNW loading. Both tensile strength and elongation at break reached their maximum when the loading percent of BCNW reached 7 wt%. Any further increase in nanowhiskers loading caused a decline in mechanical properties. The enhancement of tensile strength and elastic modulus at the expense of elongation at break up by adding small quantities of cellulose nanofibers (0.5−5 w%) was previously reported by Dufresne et al. (2014) and Dahman et al. (2014, 2016 and 2017). However, higher loading amounts of cellulose nanofibers over 5 w% could lead to miscibility and agglomeration problems that deteriorate the mechanical properties of the nanocomposites [39,40,41,42].

Figure 11 compares the effect of adding 7 wt% of BCNW to the matrices of homo-PL, TB3, and TBPU-3. As revealed in the figure, upon adding BCNW to the homo-PL matrix, both tensile strength and elongation of the obtained nanocomposite were reduced to 20 MPa and 2.0%, respectively, compared to pure homo-PL. However, adding BCNW to TB3 and TBPU-3 surprisingly resulted in significant improvements for both tensile strength and % elongation than in the free polymers. For instance, adding 7 wt% of BCNW to TBPU-3 leads to an increase of 16.5% and 58% in tensile strength and percent elongation, respectively. This is equivalent to an increase of 330% in % elongation when compared with homo-PL. These observations attributed to the poor interfacial adhesion between the hydrophilic BCNW and the hydrophobic PL matrix, where the existence of the BCNW causes an obstruction, separates the molecular chains of PL, and weaken the force of interaction among them. This in turn resulted in very weak load transfer between BCNW and PL matrix. On the contrary, the presence of PEG segments in TB copolymers and TBPUs chains helped improve the mechanical properties of the composites, where PEG segments acted as a compatibilizer. This in turn successfully improved the interaction between the polymer chains and the hydrophilic BCNW. Moreover, the hydrophilicity of PEG segments aided in the formation of strong network structure by reinforcing TBPU-3 by preventing the aggregation of BCNW and enhancing its homogeneous spreading within the entire polymer matrix in a large quantity reaching 7 wt%.

## 4. Conclusions

In this study, a total of six green and biodegradable TBPUs samples composed of PL, PEG, and PCL segments, and BDI as a nontoxic chain linker were successfully obtained and differentiated into two extremes. The low molecular weight polyurethanes with higher PEG content and shorter PL segments showed low mechanical strength and increased hydrophilicity, giving them an advantage for utilization in drug delivery and MRI imaging applications, whereas the other group that had longer PL segments and lower PEG content showed high mechanical strength and improved hydrophilicity, making them a great candidate for utilization in soft bone tissue regeneration. A promising member from the latter group (i.e., TBPU-3) was chosen as a pilot for developing five different TBPU/BCNW nanocomposites. The maximum enhancement of tensile strength and percent elongation for the nanocomposites was achieved at 7 wt% nanowhiskers loading. Introducing PEG segments in TBPUs chains demonstrated high potential for improving the interaction between the polymer chains and BCNW, keeping the polymer integrity intact and supporting a strong network structure.

## Data Availability

The data in this paper are shown in the graphs in the paper.

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
