# Peer review of "Green Biodegradable Polylactide-Based Polyurethane Triblock Copolymers Reinforced with Cellulose Nanowhiskers"

_jfb, 2023, doi:10.3390/jfb14030118_

Round 1

Reviewer 1 Report

The article on "Green Biodegradable Polylactide Based Polyurethane Triblock Copolymers Reinforced with Cellulose Nanowhiskers" appears to be a great work done by the authors.  Even though all results were explained very well, I felt there is a need to explain why "the low molecular weight polyurethanes with 678 higher PEG content and shorter PL segments showed low mechanical strength and in- 679 creased hydrophilicity, giving them an advantage for utilization in drug delivery and MRI 680 imaging applications".  It is good to have an explanation/ discussion part to made it more interesting and simple to the readers, who are not particularly from the same background.

Author Response

Re: Revised Manuscript _ jfb_2202440

Special Issue "Functionalized Polymeric Biomaterials: Design and Applications"

Reviewer 1

The article on "Green Biodegradable Polylactide Based Polyurethane Triblock Copolymers Reinforced with Cellulose Nanowhiskers" appears to be a great work done by the authors.  Even though all results were explained very well,

I felt there is a need to explain why "the low molecular weight polyurethanes with 678 higher PEG content and shorter PL segments showed low mechanical strength and 679 increased hydrophilicity, giving them an advantage for utilization in drug delivery and MRI 680 imaging applications".  It is good to have an explanation/ discussion part to made it more interesting and simple to the readers, who are not particularly from the same background.

Detailed explanation and discussion are given under section 3.4: “Degradation and Associated Morphological Changes”; see Lines 483 to 516.

Reviewer 2 Report

The article presents interesting studies on a composite with a matrix of a copolymer obtained from lactide, polyethylene glycol, polycaprolctone diol and 1,4-butane diisocyanate and cellulose nanowhiskers. These biodegradable and non-toxic materials can be used as bone cement or in medicinal applications. The form of the introduction, the description of the materials and the description of the subsequent stages of the production process of copolymers and their composites do not raise any objections. The methodology and research methodology were correctly selected. Most of the analysis of research results and their interpretation do not raise any objections.  The form of the introduction, the description of the materials and the description of the subsequent stages of the production process of copolymers and their composites do not raise any objections. The authors correctly selected the research methodology and methodology. Most analyzes of research results and their interpretation do not raise any objections. However, I have some minor comments about the way the article is presented. In line 12 - there should be an explanation of the symbol used in the name of the copolymer (PL) In several places, references to literature are presented in two forms, e.g. in verses 38, 51, 55, 145, 647 It is worth thinking about the form of representation of Eq.7 and 8 An unnumbered equation appears on line 446. BCNW assay 7 wt% rather than 7 wt%. There is no item 19 in the literature list. Substantive remarks: 1. Did the authors perform the determination of hydroxyl groups in TB and isocyanate groups in TB-BDI in order to precisely describe the composition of the obtained material. 2. In line 590, the authors stated that….Nevertheless, no Tg peaks 590 were observed in DSC curves of TBPUs related to hard or soft segments. In DSC thermographs, Tg is a characteristic inflection of PLA has a glass transition temperature between 50 and 80 °C depending on the amount of residual monomer. However, for PEG - Tg varies in the temperature range of 50 - 60 °C, depending on the molecular weight, and PCL having a glass transition temperature of -60°C. It seems that the TBPU thermographs show inflections characteristic for Tg in the temperature range of 0 - 60ï‚° C. It is worth analyzing these curves in more detail. 3. It is worth presenting an explanation why the properties of composites decrease after the introduction of wt. 8%. BCNW.

Author Response

Re: Revised Manuscript _ jfb_2202440

Special Issue "Functionalized Polymeric Biomaterials: Design and Applications"

Reviewer 2

In line 12 - there should be an explanation of the symbol used in the name of the copolymer (PL) In several places. Copolymers were explained clearly starting in line 82 (in the revised manuscript) as:

The structure of these designated polyurethanes is based on triblock copolymers (TB) that are composed of polylactide and polyethylene glycol (PL-PEG-PL) as hard segments and polycaprolactone (PCL) as soft segments. The newly synthesized TBPUs are expected to have improved hydrophilicity, biodegradation rate, and cell attachment abilities over their corresponding pure homopolymers, i.e., PLA and PCL. During the synthesis of the triblock copolymer, PEG was introduced in different ratios during polymerization reaction to enhance the hydrophilicity and degradability of PL segments [16], whereas PCL segments are used to improve flexibility and elongation.

references to literature are presented in two forms, e.g. in verses 38, 51, 55, 145, 647. Reference citation within the text was reviewed and modifications were done whenever required. Format followed for references was following Journal’s format requirements.

It is worth thinking about the form of representation of Eq.7 and 8. An unnumbered equation appears on line 446. Due to the simplicity of the equations, authors feel that the way they were represented is sufficient for readers to understand the objective of utilizing them. Equation on line 446 was numbered as Eqn 5.

BCNW assay 7 w% rather than 7 wt%. All w% is now replaced with the wt%.

There is no item 19 in the literature list. Reference 19 was added to the revised manuscript and cited in the text (19. Al-Abdallah, W., Dahman, Y. (2013) Production of green biocellulose nanofibers by Gluconacetobacter xylinus through utilizing the renewable resources of agriculture residues. J. Chem. Technol. Biol., 36, 1735–1743).

Substantive remarks: 

  1. Did the authors perform the determination of hydroxyl groups in TB and isocyanate groups in TB-BDI in order to precisely describe the composition of the obtained material. 

Authors believe that the composition has been already precisely characterized using FTIR. As stated in Line 337: “the characteristic broad peak due to hydroxyl group stretching in the TB copolymer disappeared due to urethane condensation reaction of TB copolymer with BDI, while new peaks arose at 3473 cm-1 due to −C=O−NH− stretching in secondary amide, and −N=C=O stretching at 2287 cm-1 due to isocyanate [27].”

  1. In line 590, the authors stated that…. Nevertheless, no Tg peaks 590 were observed in DSC curves of TBPUs related to hard or soft segments. In DSC thermographs, Tg is a characteristic inflection of PLA has a glass transition temperature between 50 and 80 °C depending on the amount of residual monomer. However, for PEG - Tg varies in the temperature range of 50 - 60 °C, depending on the molecular weight, and PCL having a glass transition temperature of -60°C. It seems that the TBPU thermographs show inflections characteristic for Tg in the temperature range of 0 - 60° C. It is worth analyzing these curves in more detail. 

Detained explanation is provided under section 3.5 “Thermal properties of TBPUs copolymers”; see Lines 553 to 579. 

  1. It is worth presenting an explanation why the properties of composites decrease after the introduction of wt. 8%. BCNW. Interpretation was presented in the original manuscript; according to Line 620, authors stated: ” The enhancement of tensile strength and elastic modulus at the expense of elongation at break up by adding small quantities of cellulose nanofibers (0.5-5 w%) was previously reported (Mariano, Kissi & Dufresne, 2014). However, higher loading amounts of cellulose nanofibers over 5 w% could lead to miscibility problems and deteriorate the mechanical properties of the nanocomposites [39].” In the revised, authors explained the reason of reduced mechanical properties at higher BCNW loading and cited several references: References 39, 40, 41, 42.

Reviewer 3 Report

The authors need to address the following issues before it can be accepted for publication:

1. The authors have prepared bacterial cellulose nanowhiskers and have not included any characterization study that confirms the presence of the same.

2. In Line 53 the authors have mentioned SPU. Please clarify.

3. In Line 225 the authors have mentioned H-NMR for proton NMR, the superscript 1 before H is missing.

4. Including a images of the synthesized copolymers and the polymeric films would be more beneficial for the readers

Author Response

Re: Revised Manuscript _ jfb_2202440

Special Issue "Functionalized Polymeric Biomaterials: Design and Applications"

Reviewer 3

The authors need to address the following issues before it can be accepted for publication:

  1. The authors have prepared bacterial cellulose nanowhiskers and have not included any characterization study that confirms the presence of the same. Properties of BC were characterized in previously published articles that were cited. BCNF wt% was quantified gravimetrically as stated in the article. See line 138; “Production of BC was quantified gravimetrically based on the dry weight of the BC obtained.”
  2. In Line 53 the authors have mentioned SPU. Please clarify. In Line 45,this abbreviation was defined as “Segmented polyurethanes (SPU)
  3. In Line 225 the authors have mentioned H-NMR for proton NMR, the superscript 1 before H is missing. This has been corrected.
  4. Including images of the synthesized copolymers and the polymeric films would be more beneficial for the readers. Authors think that the manuscript contains 11 Figures of scientific findings, so pictures of synthesized polymers may not add much to results. Authors will wait to see feedback from the editor if this is requested.